# Barriers and Facilitators to the Adoption and Sustained Use of Cleaner Fuels in Southwest Cameroon: Situating ‘Lay’ Knowledge within Evidence-Based Policy and Practice

**DOI:** 10.3390/ijerph16234702

**Published:** 2019-11-26

**Authors:** Debbi Stanistreet, Lirije Hyseni, Elisa Puzzolo, James Higgerson, Sara Ronzi, Rachel Anderson de Cuevas, Oluwakorede Adekoje, Nigel Bruce, Bertrand Mbatchou Ngahane, Daniel Pope

**Affiliations:** 1Department of Epidemiology and Public Health, Royal College Surgeons Ireland, Dublin D02 YN77, Ireland; OluwakoredeAdekoje@rcsi.ie; 2Department of Public Health and Policy, University of Liverpool, Liverpool L69 3BX, UK; l.hyseni@liverpool.ac.uk (L.H.); elisa.puzzolo@glpgp.org (E.P.); S.Ronzi@liverpool.ac.uk (S.R.); nigelbruce16@outlook.com (N.B.); danpope@liverpool.ac.uk (D.P.); 3Department of Nursing, Midwifery and Social Work, University of Manchester, Manchester M13 9PL, UK; james.higgerson@manchester.ac.uk; 4Internal Medicine Service, Douala General Hospital, Douala 4108, Cameroon; mbatchou.ngahane@yahoo.com

**Keywords:** household air pollution, liquefied petroleum gas, LPG, Cameroon, household perspectives, clean fuels

## Abstract

Approximately four million people die each year in low- and middle-income countries from household air pollution (HAP) due to inefficient cooking with solid fuels. Liquid Petroleum Gas (LPG) offers a clean energy option in the transition towards renewable energy. This qualitative study explored lay knowledge of barriers and facilitators to scaling up clean fuels in Cameroon, informed by Quinn et al.’s Logic Model. The model has five domains and we focused on the user and community needs domain, reporting the findings of 28 semi-structured interviews (SSIs) and four focus group discussions (FGDs) that explored the reasons behind fuel use choices. The findings suggest that affordability, safety, convenience, and awareness of health issues are all important influences on decision making to the adoption and sustained use of LPG, with affordability being the most critical issue. We also found the ability of clean fuels to meet cooking needs to be central to decision-making, rather than an aspect of convenience, as the logic model suggests. Local communities provide important insights into the barriers and facilitators to using clean fuels. We adapt Quinn et al.’s logic model accordingly, giving more weight to lay knowledge so that it is better positioned to inform policy development.

## 1. Introduction

### 1.1. Household Air Pollution: A Global Problem

Household air pollution (HAP) generated from the incomplete combustion of solid fuels used on inefficient stoves or open (three stone) fires (Figure 1a) causes close to four million deaths each year [1]. HAP has been causally related to pneumonia in children and cardiovascular and chronic obstructive pulmonary disease, and lung cancer in adults [2]. In addition, there is good evidence highlighting its role in the development of cataracts [3,4], and adverse pregnancy outcomes [5]. The health of women and children is disproportionately affected as they are generally responsible for cooking or they spend time near the fire [6,7,8,9,10] as well as collecting firewood. They are also at the greatest risk of burns from cooking on traditional stoves [11]. Furthermore, relying on polluting fuels (mostly wood and charcoal) for household energy impacts the environment through climate active pollutants of HAP and deforestation [12,13,14].

### 1.2. Transition from Solid to Clean Fuels

Low- and middle-income countries (LMICs) face a considerable challenge in securing population access to clean and efficient energy. Until quite recently, preventive actions focused on burning solid fuels more cleanly through improved biomass cookstoves (ICS) or fan-assisted ‘advanced’ biomass cookstoves (ACS) to replace traditional household cooking practices [6,15,16]. However, the latest WHO Indoor Air Quality Guidelines (IAQG) [17] and subsequent studies [18,19,20], have demonstrated that the potential of ICS/ACS to reduce harmful emissions falls by far short of the levels needed to benefit human health [21]. In addition, where cleaner fuels are available, stove stacking (the continued use of the traditional stove alongside a cleaner stove) is a common practice [22]. Therefore, there is an urgent need to scale up the use of cleaner burning fuels and energy sources.

In the absence of the widespread availability of natural gas and electricity for cooking and heating from fully renewable sources in the majority of LMICs, Liquid Petroleum Gas (LPG) represents a widely available and rapidly scalable clean burning energy option as countries transition towards alternative clean fuels such as renewable electricity [23,24]. LPG is currently used for some cooking tasks by over 2.5 billion people in LMICs (See Figure 1b and 1c for examples) [24]. However, evidence suggests that a range of factors affect the adoption and sustained use of LPG, such as affordability, access to and availability of cylinders, cooking preferences and safety concerns [25,26]. Factors which have been found to facilitate LPG uptake in resource poor contexts include government support, regulation and enforcement for safe market expansion, subsidies and financial incentives, reliability of supply and distribution networks, as well as user knowledge of improved health and cleanliness [27,28,29]. Thus, multiple factors need addressing to enable successful population transition to cleaner fuels, requiring comprehensive local and national strategies.

Many countries across sub-Saharan Africa and Asia, including Cameroon, have set ambitious targets for scaling up LPG to meet the goal of universal energy access by 2030, (Sustainable Development Goal 7) [30]. In 2011, 18% of the Cameroonian population were using LPG as primary fuel (34.3% of urban households and 1.6% of rural households) [31]. Reliance on solid fuels, however, has been falling in recent years, from 82.4% in 2001 to 65.0% in 2014, although use remains much greater in rural areas (87.5%) compared with urban areas (36.8%) [32]. Income disparities between urban and rural populations and the better and worse off have grown in recent years, although urban poverty has reduced [33] which may restrict the purchasing power of rural residents in relation to clean energy.

Solid fuel use in the South-West region stood at 57.7% in 2014 [32]. In 2016, the Government set a national goal to expand LPG penetration to 58% of the population by 2035 on an economically sustainable basis [34,35]. A national LPG Masterplan indicated the steps needed to reach the target, requiring enhanced market regulation and an investment of €400 million to strengthen LPG storage, filling, distribution, transport and retail development across the national territory [34,35].

Cameroon produces about 20% of its national LPG consumption needs and the rest is imported [34]. The price of domestically produced and imported LPG is the same, with the price being regulated and subsidised by the Cameroonian Government to protect users from international price fluctuations and maintain a steady price [35]. The cost of a 12.5 kg cylinder refill continued to be CFA 6500 or US$ 11 (i.e., US$ 0.88/kg) as of November 2019, with no changes from previous years. The price in rural areas is usually higher than in urban areas for added transportation costs [34,35]. Cameroon enforces the branded cylinder recirculation model [35] and there are several LPG marketing companies operating across the national territory. This model guarantees better cylinder maintenance and safety but does rely on sufficient investment in cylinders to avoid cylinder shortages for individual brands. Indeed, financing throughout the supply chain needs to be firmly in place if market growth is to be sustained [36]. The Masterplan identified that the growth in the number of LPG cylinders has historically been quite slow, just managing to keep pace with population growth. Investment in cylinders was therefore among the top recommendations to sustainably expand the LPG market to serve existing and additional demand [34,37].

In tandem with national plans, the LPG Adoption in Cameroon Evaluation (LACE) studies were designed to help inform the implementation of related policies to assess the LPG market and current cooking practices and identify potential enablers and barriers to scale up LPG uptake from an end-users perspective [38]. Based in South West Cameroon, the LACE-1 study surveyed 1577 households in relation to their household energy practices and identified that the most common fuels used in rural and peri-urban communities were firewood and LPG (81% and 58%, respectively) [38]. Rural households reported gathering the majority of their firewood for free (80.7% compared with 33.4% of peri-urban homes). In addition, peri-urban homes were more likely to use a secondary fuel (81.1% compared with 51.5% of rural homes) [38]. No heating needs were identified in our study area, given the climate. This qualitative study builds on these survey findings, using qualitative methods to understand the reasons behind these fuel use patterns from the perspective of the end-users.

### 1.3. Situating Lay Knowledge within an Evidence-Based Approach to Cleaner Cooking

There has been considerable discussion within the literature regarding what constitutes ‘valid’ evidence in evidence-based health, [39,40]. In evidence-based medicine, certain types of evidence have tended to be given greater legitimacy than others. For example, systematic reviews and randomised controlled trials are generally placed at the top of the evidence hierarchy, with experiential lay knowledge appearing at the bottom, if at all [41,42]. In contrast, the wider health promotion literature has critiqued the practice of privileging expert and experimental knowledge over experiential and local knowledge [43]. In this context, Lacey, Tukes et al. [44] argued that lay knowledge tends to be used as a tool to contextualise interventions at the delivery stage, rather than being given value as a source of practical knowledge to inform intervention development [44]. This is because ‘evidence based’ practice approaches are generally underpinned by a positivist worldview [45], which runs counter to a participatory approach to research and knowledge creation that views users as collaborators as opposed to subjects. Such positivist approaches fail to sufficiently recognise that narratives of user experience offer important explanations for what people do and why they do it [46].

Thus, in a cleaner cooking context, a key consideration in the design of any cleaner cooking programme should be whether an intervention is likely to meet the cooking needs of the user [47], since, if it does not, the consumer is unlikely to use it at all, let alone exclusively. In addition, users are well positioned to identify enablers and barriers in relation to certain practices and to contribute to identifying solutions to issues in the context of evaluating new interventions.

However, in the clean cooking field, where lay knowledge is (or should be) positioned in relation to evidence-based practice is not clear. There are numerous examples of qualitative studies exploring clean cooking in respect of user and community needs and perceptions [29,48,49], but little evidence to suggest that the findings these studies provide are incorporated in stove design or programme implementation.

### 1.4. A Logic Model for Scale up of Clean Household Energy

Rosenthal et al. developed a ‘logic model’ outlining the factors necessary to accommodate effective scale up of LPG [50], which has since been adapted by Quinn, Bruce et al. [51] for scaling up clean household energy more widely. It highlights five key domains or areas of influence, including a domain focused on user and community perceptions, as illustrated in Figure 2.

The aim of this paper was to explore user perspectives on the use of LPG for cooking in the anglophone Southwest region of Cameroon and to assess the capacity of the model described above [51] to identify user and community need.

## 2. Methods

The LACE studies were launched in 2016 by the University of Liverpool, UK, to identify factors affecting the adoption and sustained use of LPG and to test interventions to support communities in favouring uptake. The study design included an initial survey of cooking practices among 1577 households, the results of which are reported in Pope et al., [38]. Subsequently, the qualitative study reported here was carried out as part of LACE-1, consisting of individual semi-structured interviews (SSIs) with primary cooks and focus group discussions (FGDs) with community members. These interviews were conducted in peri-urban areas on the outskirts of a main city (Limbe, urban) and rural (villages near Buea) (Figure 3) with a purposively selected sub-set of households to obtain a greater in-depth understanding of the reasons behind current fuel use patterns. Both the SSIs and the FGDs explored views on health effects, household decision-making on fuel use, patterns of cooking, increasing uptake and sustained use of LPG, and financing options.

The epistemology of the study was informed by pragmatism [53]. The emphasis on user experience (in relation to clean cooking) and the focus on application of findings in the field (the practical implications of user perspectives) suggest that pragmatism is well suited to this study. In addition, pragmatism views knowledge as being relative as opposed to absolute and suggests that there may be singular or multiple realities. This is key in the complex field of HAP, as it accommodates multiple perspectives rather than seeking to identify one single view point as being correct [54].

The SSIs explored the views of 28 primary cooks from local households sampled purposively from LACE participants taking part in the quantitative surveys to represent three cooking fuel groups in a mix of peri-urban and rural settings: (1) exclusive or primary LPG users, (2) mixed biomass and LPG users and (3) exclusive biomass users. Primary users of LPG were defined as users who generally used LPG more than biomass but were not exclusive users. The SSIs explored individual cooking practices and the meanings that users attached to those practices. Survey participants who did not participate in SSIs were eligible to take part in FGDs. We purposively selected participants to include a range of families without children with children under five and with children over the age of five. Two FGDs took place in two rural communities of Buea (Bonjongo and Boana) and in a peri-urban community of Limbe (Mile 4), respectively. The focus groups enabled discussion among members of the community with diverse cooking experiences and fuel choices.

Ethical approval was granted by the Cameroonian ethics committee (Comité National D’Ethique de la Recherche pour la Santé Humaine) in January 2016, followed by approval from the University of Liverpool’s ethics committee and the Institutional Review Board at the Center for Disease Control and Prevention, USA, that sponsored this research. Informed written consent was obtained from participants prior to the interviews and FGDs.

### Data Collection, Management and Analysis

Fieldworkers attended a three-day intensive training course on qualitative research methods facilitated by the research team (DS and LH), which encompassed the theory and practice of qualitative data collection, the use of research tools and data transcription. Pilot interviews and FGDs were conducted in the community during the training. Online training updates were also provided during the data collection period.

The SSIs were completed first, followed by FGDs. They were conducted in English or the local language (Pidgin), and recordings were transcribed and translated verbatim to English. The field research coordinator checked transcripts for accuracy prior to commencing analysis. Interviews and FGDs were analysed by a qualitative researcher (LH) supported by web based qualitative analysis software Dedoose (www.dedoose.com) and preliminary findings were discussed regularly with the lead qualitative researcher (DS) as the analysis developed.

The focus group findings were analysed separately and then subsequently combined with the semi-structured interview findings. A thematic approach was taken to the analysis with six distinct phases of analysis being used [55]. Braun and Clarke offered a clear and systematic approach to analysing data thematically. The steps include (1) Becoming familiar with the data, (2) Generating initial codes, (3) Searching for themes, (4) Reviewing themes, (5) Defining themes, and (6) Writing up. The SSIs and FGD data were initially analysed separately and then the findings were brought together to identify common themes between the two.

## 3. Results

In order to evaluate how well the logic model represents user and community perceptions, we discuss our findings mainly in relation to the five aspects highlighted under that domain in Quinn et al.’s Logic Model (perceived affordability, perceived safety, perceived convenience, awareness and perceived prestige).

### 3.1. Demographic Characteristics

Primary cooks taking part in the SSIs (*n* = 28) were aged between 23 and 73 years old (Table 1) with the majority being educated to primary or secondary school level. Twelve participants were living in the peri-urban area with the remainder living rurally. Approximately half were married (*n* = 16). The head of household’s occupation included running a business, farming, teaching, being a student and being retired. Income level and house ownership status were similar across the three fuel groups. FGD participants had a similar demographic profile.

Overall, participants in each fuel group were split equally between rural and peri-urban communities. Similar views were expressed regarding LPG use for cooking between these communities in most domains, including regarding perceived affordability, perceptions of LPG saving schemes, perceived safety, perceived convenience, perceived awareness and perceived prestige. The key differences found were in relation to access to LPG, which is outlined in Section 3.4 and access to free firewood outlined in Section 3.6.

### 3.2. Perceived Affordability of LPG Fuel and Cookstoves

All the participants reported that affordability was one of the main barriers to LPG adoption and sustained use, especially for those without a steady income. Biomass users suggested that they were unable to afford the initial purchase of the LPG kit, whereas mixed users and even some primary LPG users said that they struggled to afford regular gas refills, preventing exclusive LPG use.

Women compared the building of a three-stone fire (locally referred to as a fireside) at no cost, to the substantial monetary outlay required to buy a gas stove. Those who already had an LPG stove and bottle costed it between 13,000 and 70,000 Central African Francs (CFA) (approx. US$ 22–120) depending on the number of burners and stove quality. Most households earn between 26,000 and 50,000 CFA (US$ 43–83 each month), with four households living on less than 25,000 CFA and only one household earning more than 100,000 CFA. Therefore, even the cheapest stove and bottle would cost a substantial part of the monthly family income. Unsurprisingly, the majority of users stressed that having a secure and sufficient income was an important factor in being able to save up to purchase the initial equipment and subsequent gas refills.
“My reason [for not adopting LPG] is that, there is no money to buy the gas. So that’s why I prefer the […] fireside.”(SSI male, age 59, biomass user, Buea, rural) 

Most primary LPG users suggested that they had to financially plan in order to be able to purchase their gas stove, through financial assistance from parents, saving up, drawing on a pension fund, paid work, or their husband’s salary. Many LPG users also felt that they were paying a poverty premium because they were constrained to buy a cheaper stove model with a shorter life and yet having to pay more in the longer term through the cost of replacement.
“…The gas stoves for sale today spoil quickly and you’d have to buy a new one tomorrow. Even though the higher quality stoves are still around, the cheaper ones are all we can afford for now. The price of a good gas stove could be about twice the cost of the cheaper ones, so people prefer to buy these Chinese light quality stoves, but the next day they end up spoiled.”(FGD 2; Participant 1; Limbe, peri-urban)

In addition, the majority of LPG users often found it difficult to find the money to purchase a refill straight away and this was considered a major barrier to exclusive LPG use.
“Many […][in our community] are farmers and cannot save [money]. Nobody can afford to spend a lot of money in one go, they have to save money gradually, and that is why the whole process is slow.”(FGD 2; Participant 2; Limbe, peri-urban)

Some participants reported paying for firewood, estimating the cost of wood fuel for one day to be around 200 CFA (US$ 0.4), although this varied according to household size. When comparing the cost of biomass to LPG, some participants were aware that over time, the overall cost of firewood was equal to, or sometimes a little bit more expensive than a 12.5 kg gas refill, yet they still regarded refills as unaffordable for many people. In practice, it was more feasible to pay an average of CFA 250 a day than to save up enough for a refill (typically costing CF 6500/US$ 11 but varying according to availability) every few weeks. Some participants suggested that the issue of cost could be addressed through reducing or further subsidising the cost of gas refills. LPG marketing companies in Cameroon do also offer smaller cylinders in the size of 6 kg (at half the cost of the larger refill) but generally, Cameroonian households continue to prefer the 12.5 kg cylinders over the 6 kg cylinders, for which sales are limited.
“With the wood, when you buy it, it is costly because in a month if you want to run the cost of your wood every day, it’s more costly than the gas.”(SSI Female, Age 24, Mixed user, Limbe, peri-urban)
“It’s difficult for people to save up 2000 CFA each week because some people cannot even afford two square meals. […] In my case now, I don’t use gas because I don’t have 6000 CFA to get a new refill, but I can easily afford 100 CFA or 500 CFA for wood or kerosene. So if they could make it in such a way that with 1000 CFA, 2000 CFA or 3000 CFA you can get a refill, it would be nice.”(FGD 1; Participant 8; Limbe, peri-urban)

Affordability also influenced choice of which foods to cook on which stove, as some participants cooked hard, slow cooking foods on the three-stone fires to preserve gas due to the need for extended cooking times.

### 3.3. Perceptions of Savings Schemes

Some biomass users felt that they were able to save up a small amount of money but also talked about the struggle to meet competing financial priorities, including the cost of their children’s education. Income levels among LPG users were similar to other fuel users, yet few LPG users reported having given anything up to purchase their gas equipment. When asked about knowledge of community-based saving schemes, many knew of ‘Njangis’ and reported that Njangis could be used to save up for a gas stove (among other items). However, only a minority of LPG users had purchased their equipment through a community savings scheme. Interestingly, the majority of biomass users suggested that they would consider participating in a loan scheme to purchase gas equipment if it were made available. However, they also voiced reservations about interest rates and the length of time available to repay the loan in instalments.
“We have not got LPG yet. The reason why is because of the cost of the children’s schooling is too much. But if God blesses, next year.”(SSI Female, Age 50, Biomass user, Buea, rural)
“I was paying my small Njangi and when it was my turn to pick, I used it to buy my gas. I had 55,000FRS from my savings, so I took the money to the market in Limbe and I bought my gas [equipment].”(FGD 4; Participant 1; Buea, rural)

### 3.4. Distance and Scarcity of Refills

LPG users found it to be a real hindrance if their cylinder brand was not available for exchange at the shop. Since, in Cameroon, cylinders are not interchangeable, participants generally had to travel further to obtain the refill from their particular band, adding transport costs. In addition, they reported that retailers can set their own prices for the gas refills and as a result, cylinder shortages drive gas prices up. Some participants suggested that this could be better managed at a government level to ensure a more steady supply of cylinders and less variation in prices. Participants living in the rural communities generally had to travel further to obtain the refills in comparison with those living in the peri-urban areas. This was perceived as a major barrier as the cost of transport to a retail point can be disproportionate and one participant noted that it can even be more than the cost of a refill. Participants would therefore like to see cylinders being more readily available through new retail points. If there were more decentralised cylinder points closer to users’ homes, that stocked a variety of different branded cylinders, this would aid users in obtaining gas in a more timely and efficient manner.
“The problem with gas being scarce is that some people sell [gas refills] […] at their own cost and not even the company price. Like sometimes I’ve wanted to go and buy gas, and one woman said, ‘Will you pay 8500frs or 9500frs for it?’ I said that I would not buy it. At that time, I decided to use my wood, another means of cooking.”(FGD 2; Participant 7; Limbe, peri-urban)
“They have to make [gas refills] readily available to people, so that one will not have to go from Mapanja to Mile 4 (~6–7km) just to get a refill which they might end up not having. This up and down travel involves paying transportation fares which may be more costly at times than the gas itself.”(FGD 4; Participant 5; Buea, rural)

### 3.5. Perceived Safety

Safety concerns were important obstacles to LPG adoption for all three user groups. Women had experienced or were aware of safety breaches and accidents that had taken place. One participant’s stove had caught fire because the hose was loose. In another recent incident, children had died in a fire caused by a gas explosion. In addition, female community members were concerned about children playing with or near the gas stove.
“It’s like a month ago, we had that fire broke out here which killed about two or three children because of gas. It’s because the gas bottle wasn’t locked, it had exploded, and the children were sleeping, and they were caught in the fire. […] You could be outside knowing that you have locked the bottle, but the child will come and operate it when you are not there and something could happen.”(FGD 3; Participant 5; Limbe, peri-urban)

Despite these concerns, LPG users were also confident that with care, gas could be used safely. Suggestions to improve safety included providing safety training and fitting cylinders with a safety valve to avoid explosions.
“Also, the gas company could provide us with safety valves like I saw on TV, that it has a kind of automatic lock that prevents leakage, hence no fire. So, if this type of security measure is taken, then it will be good.”(FGD 4; Participant 4; Buea, rural)

### 3.6. Perceived Convenience

Firewood was readily available for many biomass users at no cost, particularly among those living in rural areas, given their proximity to farms and forests. When participants compared gathering free fuel to the cost of LPG, they cited this as a barrier to investing in LPG. At the same time, women also recognised that collecting firewood (mainly done by women and children) could take several hours a day and result in back ache and other health problems. They also pointed out that cooking with wood during the rainy season was an additional burden. In addition, some communities did not have convenient access to firewood and participants suggested that this could act as a motivating factor for converting to LPG.
“What I know is that we are still developing, and we don’t need to indulge ourselves so much in gas because there is still firewood around here. This gas I say, is a luxury at the moment in this community. Many people move to gas because there is no firewood.”(FGD 2; Participant 2; Limbe, peri-urban)

In terms of stove convenience, participants clearly articulated the relative benefits and disadvantages of LPG and traditional stoves. The LPG stove was considered clean, very fast, and some had multiple burners, therefore, were convenient when cooking large meals. These were all seen as distinct advantages. Conversely, the three-stone fire was described as dirty, took time to light and heat up, and only had one burning source, meaning that only one dish could be cooked at once. However, women valued its ability to accommodate large pots, its speed once the fire was well established and for cooking food that needs to simmer for a long period. Mixed users reported regularly stove stacking (using both the LPG stove and the three stone fire for different purposes) mainly to economise on gas and also for safety reasons, such as when children were in the home. LPG was preferred for reheating food and cooking foods such as rice, spaghetti, boiled cocoyam and frying eggs. In contrast, ‘hard’ foods that needed time to cook including Fufu, Eru, Koki, beans, corn-chaff and others were cooked on open fires to preserve gas, suggesting that the LPG stove did not in fact meet all their cooking needs.
“When I have to cook certain things like when somebody is having a special occasion and asks me to help cook some of the meals and the meals are heavy, I prefer to do that on the traditional stove, I can’t use my gas.”(SSI Female, Age 25, Mixed User, Buea, rural)
“The three-stone fire is easy for some dishes, but the gas cooker makes your cooking clean and is less stressful.”(SSI Male, Age 36, Mixed user, Limbe, peri-urban)

LPG users found that it very inconvenient when the cylinder ran out whilst cooking, as they had to switch fuels, since most households can only afford to purchase one cylinder. This could result in having to light the three stone fire in the middle of cooking a meal, or seek a refill if the money is available, or sometimes even waiting days until there is enough money to buy a replacement. Some participants suggested the need for an indicator on cylinders to signal when the gas is low so they can be prepared to purchase a new refill.
“Very frustrating. Sometimes gas can finish while I am cooking, I will have to stop and helter-skelter to look for gas. If I don’t have money at that time to get the gas I can ask to take my pot to a neighbour’s house so that I can complete my cooking. But if I have the money, I have to start carrying the bottle up and down to get gas (…) the same day or the next day because I cook solely with gas.”(SSI Male, Age 38, LPG user, Buea, rural)
“In my experience, you can’t determine when the gas will run out. You can put a pot on the stove and the gas can run out without warning. When you buy your gas, it should have a meter that indicates when it’s getting finished or a line that indicates the fullness of the gas bottle.”(FGD 1; Participant 8; Limbe, peri-urban)

### 3.7. Perceived Awareness

Health considerations were reported as an important factor for biomass users wanting to change to LPG. Most reported experiencing a variety of health problems, including chest pain, breathing difficulties, eye problems, cough, runny nose and headaches, and most voiced their concerns about this, attributing the symptoms to the smoke. Because of this, some participants did not allow their children to come near the fire. In contrast, LPG users did not report experiencing negative health effects from cooking and mixed users noted an improvement in their symptoms when cooking with gas.
“The truth is whenever I use the LPG stove it is different from when I use the three stone fire side. With the three stone fire side, your eyes tear when you are using it and even when you are through with what you are cooking. But with the LPG stove, yours eyes are normal.”(SSI Female, aged 53, mixed user, Buea, rural)

Awareness of other advantages of using cleaner fuels was also evident and have been discussed earlier, including for example, that participants were aware that in the longer-term, LPG may be cheaper than using biomass.

### 3.8. Perceived Prestige

The issue of prestige was not widely expressed as a motivation for changing to gas, although it was mentioned by one participant.
“In my community, if you use gas to cook, you have a certain prestige amongst others; you are seen as being fortunate.”(FGD 4; Participant 8; Buea, rural)

## 4. Discussion

### 4.1. Factors Influencing the Transition from Biomass to LPG

This study outlined, from the perspective of users in peri-urban and rural communities in Southwest Cameroon, a number of barriers and facilitators to promoting the adoption and sustained use of LPG as a cooking fuel. Affordability was the most common issue, mentioned by all participants and has been documented elsewhere as an important barrier to switching as well as continued fuel use [27,29]. It is well established that where biomass can be collected for free, there is less financial incentive to switch to use cleaner fuels [56]. However, some biomass-using households in this study reported purchasing firewood and were also aware that firewood was probably more expensive to buy than gas refills in the longer term. Indeed, an increasing body of evidence shows the costs of cooking with LPG are similar if not lower, to the costs for purchased biomass, particularly charcoal [26,57,58,59,60]. However, users pointed out that biomass fuels can be purchased in smaller quantities which is much more manageable on a low income. Therefore, the issue of affordability is a practical barrier, where users need to save over a period of time to purchase the initial LPG equipment and then, the subsequent refills. Small loans (e.g., microloans) for start-up LPG kits have shown some promise in Cameroon and elsewhere [37,61,62,63,64,65]. However, credit checks, lack of knowledge of financial offers, and potentially high interest rates can unintentionally lock out non-traditional borrowers.

The previous literature has also reported safety concerns as a barrier to LPG adoption [27,66]. In this study, it was biomass users, particularly in rural areas, who raised most safety concerns regarding the use of LPG. They had concerns about the risk of gas leaks and explosions in relation to the unsafe use of LPG, or children playing with the stove. LPG users may have had concerns previously, but these seemed to lessen with experience of use, suggesting that ‘hands-on’ experience and education may be effective in reducing safety concerns. However, this needs to be accompanied by national regulation and enforcement in respect of market rules and safety practices. In Cameroon, there are already regulations in place to prevent cylinder exchange between brands to curb illegal refilling and unauthorised rebranding, but more could be done in terms of enforcing more stringent cylinder checks (and re-qualification practices). In addition, the Masterplan recommends the use of pallets during cylinder transportation and unloading operations to avoid cylinder damage [35].

Study participants considered LPG more convenient in relation to not having to gather fuel, especially wet fuel during the rainy season, speed of lighting the stove, cleanliness, ease of use and the presence of multiple burners. However, LPG was seen as not being able to accommodate large (rounded) pots and less convenient in relation to accessibility and availability of cylinder refills. Not being able to accommodate large pots could be an important factor when choosing which stove to use when cooking for a large family. Similar findings have been reported in other settings [57,67].

The only solution to resolving the supply problem is for LPG companies to significantly invest in cylinders and increase last-mile distribution with retail points within walking distance from end-users [35,36,67,68]. This was also recommended by participants in the study. Given that it is not possible to exchange between cylinder brands in Cameroon, shops would need to expand their inventories and have a consistent stock of all available cylinder brands. In addition, some participants suggested the use of a meter so that they would be able to see when gas is running low. This could be useful but would involve extra costs and maintenance, and even in developed markets, such meters are not used. Instead, having two cylinders and being able to switch from one to the other is preferred practice in some settings [57], although this would involve initially paying for two cylinder deposits instead of one.

The reasons for stove and fuel stacking identified in this study have also been reported elsewhere in relation to switching to cleaner fuels [22,35,57,69]. Multiple stove use practices are generally rational and logical responses to cleaner cooking methods either being unaffordable, unavailable, unreliable or not meeting cooking needs. This is discussed further in the following section.

### 4.2. Suitability of the Logic Model and Proposed Adaptation

The model presented by Quinn et al. [51] provides a useful means of understanding the barriers and facilitators, to scaling up of LPG from a user perspective. Overall, four of the five main categories identified by the user and community perceptions domain in the framework were very useful in framing the findings of the users’ voices in this study. As mentioned previously, Quinn et al. have addressed some of the limitations of Rosenthal et al.’s logic model [50] and expanded the user aspects. We would like to continue the development of this approach by suggesting some additional adaptations to further promote user views within the fuel scale up process.

Currently, user views lie both literally and figuratively at the base of the hierarchy. In order to promote user views as a starting point from which to evaluate whether cleaner cooking meets user needs, we advocate placing user views at the top. In addition, we recommend removing the word ‘perceptions’ from both within the title of the user domain and also in relation to four of the identified aspects. This will ensure that the weight of lay knowledge is not diminished in respect of the other domains.

Thus, affordability would be presented not as a ‘perceived’ but as a practical barrier that needs to be addressed by actions elsewhere in the logic model (through subsidies as part of an ‘enabling environment’ for example). The same is true for ‘perceived’ safety’, which is also recognised within the logic model under Dimension Two, ‘Safety regulation and practices’ without the qualification of perception. The findings of this study suggest that users are indeed aware of safety issues with LPG equipment and this issue will require addressing in other domains of the model before wider uptake and sustained use is likely to be successful.

Furthermore, issues such as non-availability of LPG refills and difficulties with accessing the refills, are again real physical barriers which need to be addressed by other domains, (e.g., industry and services level, market regulation). This involves companies investing in the supply of cylinders. However, community members could be well placed to suggest suitable local outlets or models for local distribution and supply, including the potential of community members becoming involved in last mile distribution. This would require a direct link between the ‘User and community’ domain, and that of the domain ‘Industry structure, and services’ which could be added to the model.

Cooking needs are also categorised under ‘perceived convenience’ but it could be argued that meeting cooking needs is not an issue of convenience but rather, the key issue with which to understand user cooking behavior; therefore, the model would benefit from affording it a more central position. We advocate changing the title of the user and community domain to ‘User and community cooking needs’, since all the other aspects of this domain need to ensure cooking needs are met. Furthermore, whether or not cooking needs can be met by the use of a cleaner stove/fuel can be measured in relation to the practice of stove stacking since stove stacking is a pragmatic response by users to address the barriers they identify. For example, affordability was the major reason given for stove stacking, but participants also reported stove stacking for convenience, to smoke fish, to cook for a large number of people and for safety reasons (e.g., when children around). In addition, stacking is necessary to overcome the very practical problems reported in relation to accessibility and availability of LPG refills. If users find certain practices more convenient, practical, or feasible than others, these are the practices that need to be addressed to bring about market expansion. For example, if users prefer to cook in bigger pots to feed large families, this will need to be taken into account in the design of the stoves. This is overlooked in much of the literature, but has been highlighted by Ruiz, Mercardo and Masera [22] as being a significant factor in the uptake and sustained use of cleaner fuels. Similarly, a separate aspect focusing on access and availability will allow a focus on factors affecting access from a user perspective.

Finally, in this study, ‘perceived prestige’ did not seem to be a major factor in fuel switching decisions, but this was not an issue that we specifically enquired about.

In summary, the adjustments discussed above to the logic model would emphasise the significance of user practice, giving further weight to user knowledge, enabling it to play a fuller role in market expansion alongside expert knowledge in the field of clean cooking. These are illustrated in Figure 4 below.

Within the model, whether or not a stove meets user cooking need should be a pivotal point for assessing both stove design and the effectiveness of market expansion and stove stacking should be viewed as a measure of where current scale up options fail to meet user need rather than as a practice that users need to be educated not to undertake. In this respect, it becomes a tool to evaluate whether actions being undertaken across all logic model domains promote exclusive cooking.

### 4.3. Strengths and Limitations of the Study

This qualitative study carried out in Southwest Cameroon has provided insights into cooking practices among rural and peri-urban communities that have some access to LPG for cooking. We explored the views of users who exclusively use biomass as well as users who exclusively or primarily use LPG for cooking and identified a number of barriers and facilitators to adoption and sustained use, which are common among rural and peri-urban users. Even though the findings represent only the views of households in Southwest Cameroon, the literature suggests that many of these factors are common in similar contexts throughout the world. It is therefore likely that the findings can be extrapolated to other settings at an early stage of use of LPG as a domestic fuel with limited penetration facing similar barriers. However, further research is required in different geographical settings to demonstrate transferability to other LPG market contexts.

The study also utilised and adapted a logic model for clean cooking fuels scale up and made recommendations for promoting the role of the end-user in policy making in relation to clean cooking. These recommendations now need to be applied in practice to assess whether greater emphasis on the user perspective can effectively increase demand for clean fuels, contribute to market expansion to meet that demand, and reduce stove/fuel stacking practices to achieve the greatest health, environmental and societal benefits.

### 4.4. Recommendations for the Cameroon Context

Our findings contribute to strengthening the recommendations included in the LPG for Clean Cooking Masterplan published in 2016. This masterplan presents a roadmap of actions required to sustainably and safely expand the LPG market throughout the country [34]. Our findings underline the need for expansion of last-mile distribution as a critical element in this roadmap to ensure sustained access to LPG refills among existing users as well as to encourage non-LPG users (particularly those who pay for their fuels) to adopt a cleaner cooking option. In addition, communities support the proposal for financial incentives or borrowing practices (e.g., from local culturally appropriate savings group or more established financial institutions) to be scaled up across the entire country, in order to support less affluent households to make the transition. Finally, an education campaign in the safe use of LPG, which reflects local cooking practices (including how to save on gas), could also be promoted to overcome safety barriers and encourage uptake.

## 5. Conclusion

Local communities provide valuable insights into the barriers and facilitators they face to exclusive use of clean fuels. The revised logic model for scaling up clean cooking which is described here offers an alternative approach placing the ‘user’ at the heart of the discourse in order for national, regional or local clean cooking programmes to focus on the cooking needs of local communities more effectively and to encourage more sustained use of clean fuels. We recommend giving more weight to lay knowledge, and in particular, to directly seeking to address the issues that users face in relation to their everyday cooking practices to enable their views to more effectively inform local and national policy and market practices in relation to cleaner cooking.

## Figures and Tables

**Figure 1 ijerph-16-04702-f001:**
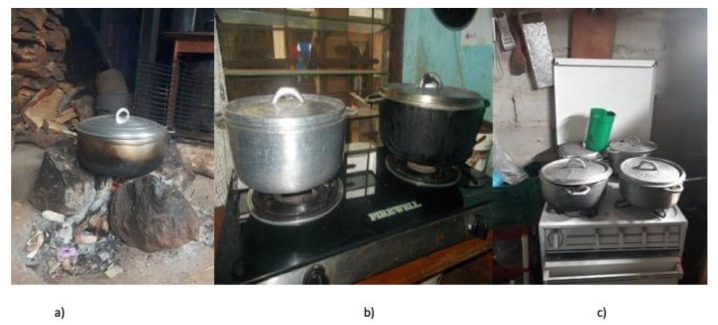
(**a**) A three-stone fire, (**b**) a two-burner Liquid Petroleum Gas (LPG) stove and (**c**) a four-burner LPG stove in South-West Cameroon. Source: LACE Studies.

**Figure 2 ijerph-16-04702-f002:**
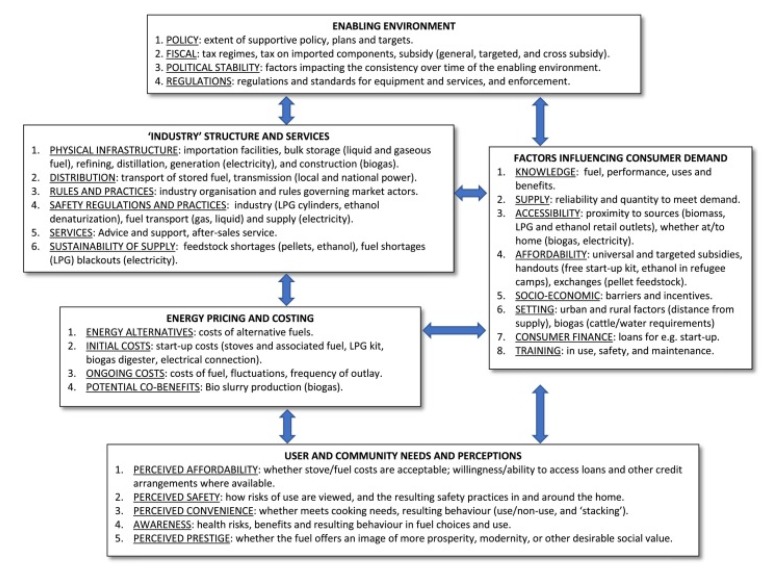
Reproduced from Quinn et al. (2018) An analysis of efforts to scale up clean household energy for cooking around the world. *Energy for Sustainable Development* (46, pp. 1–10).

**Figure 3 ijerph-16-04702-f003:**
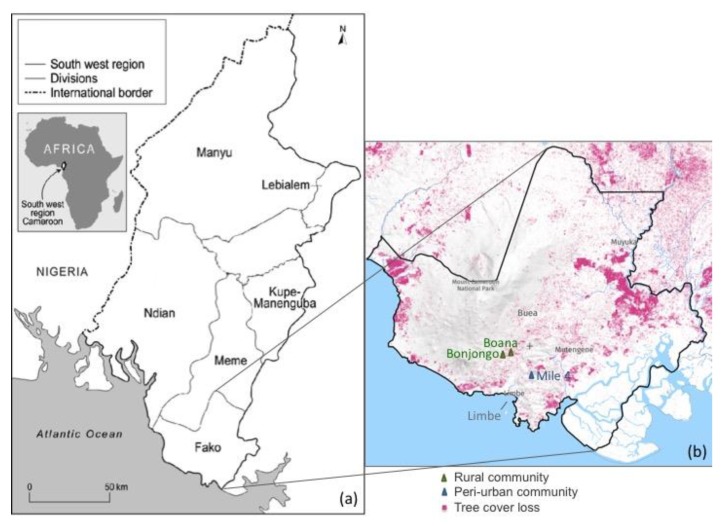
Map of Southwest Cameroon highlighting the study areas near Buea and Limbe. Reproduced from Global Forest Watch 2019. Cameroon, South Ouest, Fako. Available at: https://www.globalforestwatch.org (accessed on 18 November 2019) Note: (**a**) Southwest divisional map; (**b**) enlargement of the Fako division where the study was conducted, over-imposed with a tree cover loss map [52].

**Figure 4 ijerph-16-04702-f004:**
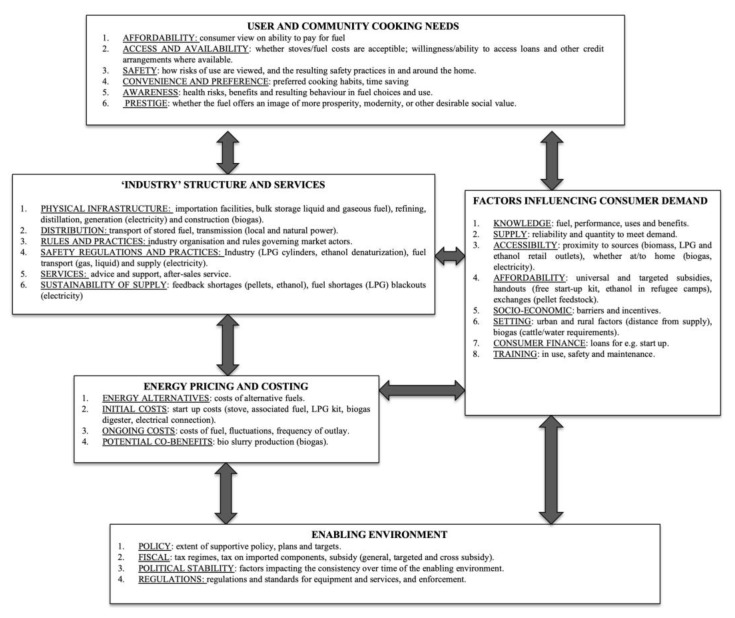
Revised generalised logic model for clean fuel scale up.

**Table 1 ijerph-16-04702-t001:** Demographic characteristics of semi-structured interviews (SSI) participants (*n* = 28).

Age Range	Females (*n*=)	Males (*n*=)	No. with Children	No. with Children under 5	No. Using LPG	No. Using Biomass	No. Using Biomass and LPG
20–40	13	3	16	11	6	3	7
41–60	5	3	8	4	1	6	1
61–75	3	1	2	2	2	2	0
Total	21	7	26	18	9	11	8

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
