# Peer review of "Barriers and Facilitators to the Adoption and Sustained Use of Cleaner Fuels in Southwest Cameroon: Situating ‘Lay’ Knowledge within Evidence-Based Policy and Practice"

_ijerph, 2019, doi:10.3390/ijerph16234702_

Round 1

Reviewer 1 Report

Barriers and facilitators to the adoption and sustained use of cleaner fuels by Stanistreet et al.,

This is a well-done research and well-written manuscript introducing an important issue- barriers and enablers in promoting and sustainable use of cleaner fuel, more specifically LPG here. I only have two concerns here:

Response and findings from the field study is given mainly by the number of respondents to each question/domain, e.g. affordability, distance, refills, … besides this qualitative information, it is interesting to know any quantitative results on these?

In field survey and in analysis of results, any scores, or something similar, to quantitatively evaluate the impacts and relative contributions of each factor?

Reviewer 2 Report

This study involves the analysis of survey and focus group data for solid fuel and LPG users in southwest Cameroon. The authors develop an updated model that can help clarify decision making by the user population based on multiple variables. This manuscript describes the improvement from previous models due to direct input from users. It is well-written and desgined.

Several of the header sections have strange numbers (i.e. Introduction has "1110") following them. These should be removed. It would be useful to include a map of the study area as many readers may not be familiar with Cameroon. Adding a photograph of a 3-stone fire (and similar) stoves would be useful for readers not familiar with these technologies. A general question - why is Limbe considered peri-urban and not urban? It seems that the first part of the manuscript describes studies on urban areas and it seems that Limbe is urban. An argument made for using wood was that it is plentiful in certain regions. Since the authors conducted experiments in multiple locations, an enhancement to the map suggested above would be to not only show the study locations, but a land use/vegetation layer to show which areas have available wood. Line 208 seems to have a typo ",,," instead of "..." Citations seem to be in "()" and not "[]" as is the IJERPH standard. Figure 2 has two capitalization anomalies (different from all others): 1) Under "Industry Structure and Services", "Services", the word "Advice" is capitalized; 2) In "Energy Pricing and Costing", "Potential Co-benefits", the word "Bioslurry" is capitalized. In Figure 2, "Affordability" at the top has no description. This needs to be better explained so it can be clearly differentiated from "Energy Pricing and Costing". Line 464: "increasing" should be "increase"

Reviewer 3 Report

The study addresses a relevant issue as users’ preferences for fuels, analyzing barriers and facilitators. This information is very important in order to be able to focus public policies on reducing these barriers and promoting facilitators. Most public policies that have not taken into account the user preferences, have failed to achieve the expected impacts on energy transition and adoption of cleaner fuels.

Introduction:

More detail on the context is required to understand the specific study case. Information on energy consumption by households should be explained to understand if firewood and gas are used only for cooking or also for heating in some months of the year. It is also important to understand the situation of household energy poverty, considering aspects as: percentage of income spent on energy, access to energy, availability of clean fuels, etc.

This information would help to contextualize this specific case study, since the introduction of the paper refers to LMICs countries. But it is important to consider that in this type of countries there are very different contexts of energy consumption, technologies and fuels; according to social, cultural, climatic and geographical particularities, among others.

It is also relevant to know if fuels are imported, price differences and other aspects that can help to better understand this particular situation.

On the other hand, the policies and programs that currently exist to encourage the adoption of LPG and the reasons why they have not had the expected results should be explained. In addition to this, the regulations on prices, quality of fuels, type of distribution and others should also be presented.

Methodology:

More detail on the methodology that was used is required. This section should include information on how the model works, how data is analyzed, how each of the factors is weighed and how the number of interviews was determined.

It is also important to know why this methodology is used and not another, for example as a choice experiment to identify the valuation of the different attributes by the users.

Results:

Rural and peri-urban communities are studied, which probably determines different possibilities of collecting biomass and accessing LPG. The analysis could show how the level of rurality of the communities affects preferences for different fuels.

The results could be presented in a table or a figure so that they are easier to understand, especially which are the perceptions that have the greatest influence on the adoption of cleaner fuels.

Discussion:

In the discussion of the results, some recommendations should be incorporated to improve existing public policies or recommendations for new policies to be implemented according to the results of the study. Recommendations regarding financial instruments (both for the purchase of equipment and for the purchase of fuel) that should be implemented and other complementary instruments such as information or other types of incentives are necessary. The results of user preferences are very relevant for these recommendations.

With the results, solutions and programs for different types of users and different needs can be identified, this could be presented in the discussion section.
